# Gastric Microbiota in *Helicobacter pylori*-Negative and -Positive Gastritis Among High Incidence of Gastric Cancer Area

**DOI:** 10.3390/cancers11040504

**Published:** 2019-04-10

**Authors:** Boldbaatar Gantuya, Hashem B. El-Serag, Takashi Matsumoto, Nadim J. Ajami, Khasag Oyuntsetseg, Dashdorj Azzaya, Tomohisa Uchida, Yoshio Yamaoka

**Affiliations:** 1Department of Environmental and Preventive Medicine, Oita University Faculty of Medicine, 1-1 Idaigaoka, Hasama-machi, Yufu-City, Oita 879-5593, Japan; medication_bg@yahoo.com (B.G.); tmatsumoto9@oita-u.ac.jp (T.M.); azzaya2000@gmail.com (D.A.); 2Department of Internal Medicine, Gastroenterology Unit, Mongolian National University of Medical Sciences, Zorig Street, Ulaanbaatar-14210, Mongolia; oyuntsetseg.kh@mnums.edu.mn; 3Department of Medicine, Gastroenterology and Hepatology Section, Baylor College of Medicine, 7200 Cambridge Street, Houston, TX 77030, USA; hasheme@bcm.edu; 4Alkek Center for Metagenomics and Microbiome Research, Department of Molecular Virology and Microbiology, Baylor College of Medicine, Houston, TX 77030, USA; nadimajami@gmail.com; 5Department of Molecular Pathology, Oita University Faculty of Medicine, 1-1 Idaigaoka, Hasama-machi, Yufu-City, Oita 879-5593, Japan; tomohisa@oita-u.ac.jp; 6Global Oita Medical Advanced Research Center for Health, 1-1 Idaigaoka, Hasama-machi, Yufu-City, Oita 879-5593, Japan

**Keywords:** *Helicobacter pylori*, non-*Helicobacter pylori*, gastritis, microbiota, 16S rRNA

## Abstract

*Helicobacter pylori* (*H. pylori*) related chronic gastritis is a well-known major etiological factor for gastric cancer development. However, *H. pylori*-negative gastritis (HpN) is not well described. We aimed to examine gastric mucosal microbiota in HpN compared to *H. pylori*-positive gastritis (HpP) and *H. pylori*-negative non-gastritis group (control). Here, we studied 11 subjects with HpN, 40 with HpP and 24 controls. We performed endoscopy with six gastric biopsies. Comparison groups were defined based on strict histological criteria for the disease and *H. pylori* diagnosis. We used 16S rRNA gene amplicon sequencing to profile the gastric microbiota according to comparison groups. These results demonstrate that the HpP group had significantly lower bacterial richness by the operational taxonomic unit (OTU) counts, and Shannon and Simpson indices as compared to HpN or controls. The linear discriminant analysis effect size analysis showed the enrichment of Firmicutes, Fusobacteria, Bacteroidetes and Actinobacteria at phylum level in the HpN group. In the age-adjusted multivariate analysis, *Streptococcus* sp. and *Haemophilus parainfluenzae* were at a significantly increased risk for HpN (odds ratio 18.9 and 12.3, respectively) based on abundance. *Treponema* sp. was uniquely found in HpN based on occurrence. In this paper, we conclude that *Streptococcus* sp., *Haemophilus parainfluenzae* and *Treponema* sp. are candidate pathogenic bacterial species for HpN. These results if confirmed may have important clinical implications.

## 1. Introduction

Most of the gastric cancer incidence is arising from East-Asian countries. Among them the leading countries are Korea, followed by Mongolian and Japan. By the mortality rate the worst country is Mongolia. It is reported that 97% of overall gastric cancer belonged to the sporadic type [1]. Gastric cancer is the multifactorial disease however the *Helicobacter pylori (H. pylori)* infection is considered a major etiological factor [2]. The development of next generation sequencing technologies allow for in depth studies of the gastric microbiota composition. A recent review paper for gastric cancer microbiota highlighted *Escherichia Shigella* and *Burkholderia* within the Proteobacteria phylum; *Lactobacillus, Lachnospiraceae, Streptococcus,* and *Veillonella* within the Firmicutes phylum; and *Prevotella* within the Bacteroidetes phylum. Although great advances have been made in understanding the complex interplay between the gastric microbiota and *H. pylori* in the development of gastric inflammation and cancer, detailed studies are still needed in well-defined human populations to compare differences in the microbiota of *H. pylori*-infected persons with and without neoplastic lesions [3].

For the gastric cancer pathogenesis, *H. pylori* plays the main role for developing gastric cancer due to atrophic gastritis, which mainly develop an intestinal type gastric cancer and non-atrophic gastritis, which mainly develop a diffuse type gastric cancer [1,4]. Gastritis is a histopathological entity that is characterized by neutrophilic infiltration. Previous studies reported that *H. pylori*–negative gastritis is a common histopathological and clinical condition that is independent from *H. pylori* gastritis in US populations [5]. A recent study showed a 24-fold increase in neutrophil counts in the gastric cancer tissue, and nine-fold increase in gastric intestinal metaplasia tissue compared to the normal gastric tissue. *H. pylori* stimulates the neutrophil accumulation in epithelial cells with the production of inflammatory mediators, reactive oxygen and nitrogen species, which contribute to the disruption of gastric epithelial function [6] and DNA damage [7]. Neutrophil infiltration is also associated with the E-cadherin downregulation, cell proliferation and gastric carcinogenesis [8]. While it is well known that gastritis is induced by the *H. pylori* infection, [9] the role of other bacterial infections of the stomach is not well studied.

The composition and diversity of microbial communities in the stomach is greatly influenced by gastric acidity [10]. A well-established factor influencing gastric acidity is the *H. pylori* infection, which triggers the development of chronic active gastritis that may subsequently progress to chronic atrophic gastritis. Previous studies using the 16S rRNA gene sequencing, reported that the gastric microbiota diversity in gastric tissue samples was dramatically reduced in *H. pylori*-positive cases compared to negative cases [11]. *H. pylori* comprised most (the average 72%) of sequencing reads among patients with positive conventional *H. pylori* tests (i.e., culture, rapid urease test, serology and histology) [12]. However, the gastritis status was not examined in the study. Another study reported a high abundance of *Streptococcus* sp. in the *H. pylori*-negative antral gastritis. However, the definition of *H. pylori* negative cases in that study was based on the rapid urease test and 16S rRNA gene sequencing, and the gastritis diagnosis was based on endoscopic findings and presence or absence of gastric atrophy but not neutrophils. Furthermore, the number of subjects examined was small (five cases and five controls) [13]. Lastly, the gastric microbiome and acidity are also modified by antibiotics, proton pump inhibitors (PPIs) and gastric surgery [14,15]. Therefore, there have been no studies of the gastric microbiome in *H. pylori*-negative gastritis that considers a larger number of subjects with careful definitions of the *H. pylori* infection, gastritis and previous medical history.

To address this important gap in the literature, we examined differences in the gastric microbiome for *H. pylori*-negative gastritis (HpN) compared to the *H. pylori*-positive gastritis (HpP) and *H. pylori*-negative non-gastritis group (control) in the Mongolian population employing rigorous definitions for cases and controls.

## 2. Results

### 2.1. Study Population

Samples were collected and sequenced for microbiome profiling from 220 subjects. Of these, 145 cases were excluded due to our criteria for defining study groups. The analysis presented here pertains to samples from a total of 75 subjects (11 HpN, 40 HpP, and 24 controls). The HpN group was defined by the positive histological diagnosis for gastritis and negative by all *H. pylori* tests including 16S rRNA gene amplicon sequencing. The HpP group was positive by the histological diagnosis for gastritis and positive by all *H. pylori* tests including 16S rRNA gene amplicon sequencing. The control group was based on the histological diagnosis that neutrophil infiltration was not seen in the gastric mucosa, Updated Sydney system score 0 (none of infiltration) for gastritis and negative by all *H. pylori* tests including 16S rRNA gene amplicon sequencing. The flow chart of the sample selection is shown in Appendix A
Appendix A. The overall mean age was 46.8 years (standard deviation 15.6), and 72% were females. There were no significant age or gender differences among the three groups (Appendix A).

### 2.2. Detection of Operational Taxonomic Units (OTUs)

We identified a total 2,550 OTUs of which 125 OTUs were filtered based on OTU low abundance (minimum total count < 10 or minimum relative abundance 0.05% of OTUs as a default option of the CLC genomic workbench 8.5.4 version for microbiome analysis). The overall count reads per sample was 6,181,354 OTU reads (minimum 2,053 and maximum 227,528), and the dataset was rarefied to 2,053 OTU reads per sample (Appendix A). The minimum total count per OTU was 41 (Fusobacterium OTU ID: 828676) and maximum was 4,513,446 OTU reads (*H. pylori* OTU ID: 10952).

The rarefaction analysis is shown for HpN (Appendix A), HpP (Appendix A) and control (Appendix A) groups, respectively. Most HpN group cases had higher bacterial diversity (richness) (Overall OTU relative abundance scaled up to 100%) but possessed lower read numbers (Scaled by raw count numbers up to 200,000). HpP group cases had a variety of richness and had generally higher read numbers per sample. The cases of control group had a variety of richness and had relatively lower read numbers per sample.

Recovered taxonomies (Appendix A) and their relative abundance are shown at genus level per sample (Appendix A). Cases with HpN and control groups had a higher bacterial diversity but were observed to have a lower read count compared to the HpP group. The HpP group had overall less diversity due to the dominance of *Helicobacter* genus as observed by an increased read count mapping to the *Helicobacter* genus. One HpN case showed a striking dominance by *Treponema*. The sequenced data will be deposited on the Genbank as Accession Number (in proceeding).

### 2.3. Microbiome Diversity

Microbial community richness and evenness as represented by Shannon (Figure 1A), richness (Figure 1B) and Simpson’s diversity indices (Figure 1C) showed statistically significant differences between the HpN vs. HpP, and for HpP vs. control, but not for the HpN vs. control group comparison. The beta-diversity analysis using Bray-Curtis distance metrics revealed differences in the microbial structure between the HpP and HpN groups (R2 0.5, *p* < 0.0003), but no significant differences between the HpN and control groups (R2 0.05, *p* = 0.2). Similarly, the Jaccard distance metric was highly significant (R2 0.47, *p* < 0.0003) for the HpN vs. HpP group comparison but not for the HpN vs. control group (R2 0.04, *p* = 0.1). The Discriminant Analysis of Principal Components (DAPC) analysis used to identify the group level clusters showed a clear separation among the three groups. The HpP group had a more distinct composition compared to the HpN or control groups, which were closer to each other in composition (Figure 1D). These differences were driven by the absence or presence of taxonomies in which 120 were determined in each group to be part of a core microbiome, control and HpP groups shared 3 taxa (*Enterococcus* sp. OTU ID: 766768, *Lactobacillus* sp. OTU ID: 324926, *Lactobacillus* sp. OTU ID: 851733), HpN and HpP shared one OTU (*Fusobacterium* sp. OTU ID: 828676) and one OTU (*Treponema* sp. OTU ID: 2707164) was unique in the HpN group (Figure 1E). Then, we described the differences of each group based on abundance.

The alpha diversity analysis showed that among the HpP group, the *H. pylori* relative abundance was around 90%, whereas bacterial taxa were more diverse among HpN and control groups. The overall relative abundance is shown for phylum level (Figure 2A) and for genus level (Figure 2B). Differences in averaged bacterial abundance were determined by analysis of variance (ANOVA). Three *Streptococcus* OTUs had significantly higher abundance in HpN than the control group (Appendix A). For the HpP vs. HpN comparison, *H. pylori* was higher in HpP whereas most of the remaining OTU taxonomies were significantly higher in the HpN group. Top 10 OTU taxonomies are shown for HpP and HpN comparison (Appendix A).

Biomarker discovery using the linear discriminant analysis effect size (LEfSe) algorithm showed that Proteobacteria was enriched in the HpP group; and Firmicutes, Bacteroidetes, Fusobacteria and Actinobacteria were enriched in the HpN group at phylum level (Figure 2C). *Helicobacter* was the only distinguishing taxon in the HpP group, whereas 14 distinct taxa distinguished HpN from the HpP and control groups. At the genus level, nine taxa distinguished the control group from the rest (Figure 2D). Using the Wilcoxon test, we found 10 species as potential biomarkers for the HpN compared with control as well as the HpP group that *Streptococcus* sp. (OTU ID: 1010458, ID: 1078207, ID: 526131, ID: 967427, ID: 989579, ID: 525966), *Haemophilus parainfluenzae* (OTU ID: 4320756), *Fusobacterium* (OTU ID: 938948), *Veillonella* (OTU ID: 511378) and *Prevotella Pallens* (OTU ID: 705241) (Appendix A).

The network analysis showed bacterial interactions among all three-comparison groups (Appendix A), the HpN and control (Appendix A), and the HpN and HpP groups (Appendix A). Similar to the overall relative abundance, *H. pylori* did not co-occur with other taxa and dominated in the HpP. High microbial co-occurrence was observed in the HpN and the control group suggestive of a richer community.

### 2.4. Univariate, Receiver Operating Characteristics (ROC) Curve and Multivariate Analysis

From the biomarker discovery analysis (Appendix A) between HpN vs. HpP and HpN vs. control, we selected the 10 candidate species (for HpN (described above)), which were consistent in both comparisons between HpN vs. HpP, and HpN vs. control group. The average abundance and statistical significance based on the *t*-test among the HpN, HpP and control groups is shown for each candidate (Figure 3).

All candidates were significantly higher in HpN than the control group followed by the HpP group. Additionally, *Treponema* (OTU ID: 2707164) was significantly higher in the HpN group compared with the HpP group by ANOVA (Figure 4A) but not by the LEfSe. This result was driven by the high abundance of *Treponema* observed in one HpN case. The arithmetic mean ± standard deviation of the relative abundance (raw count number reads) were 5.9 ± 19.5 (1736 ± 5758) for HpN, 0 ± 0 (0.02 ± 0.1) for HpP, and 0 ± 0 (0 ± 0) for controls (*p* < 0.036). Similar to the case of *H. pylori* in the HpP group, *Treponema* (OTU ID: 2707164) strongly dominated over other taxa in that sample (Figure 4B). Inflammatory cell accumulation with hemorrhagic change was observed in the gastric histopathology of such *Treponema* positive by the 16S rRNA patient (Figure 4C).

We selected the 10 candidate species that were significantly enriched in HpN comparing with each of the HpP and control group using Wilcoxon tests. In addition, *H. pylori* were included. Area under curve (AUC) was <0.7 for *Streptococcus* (OTU ID: 989579), *Prevotella* (OTU ID: 705241) and *Treponema* (OTU ID: 2707164) which were excluded. The remaining candidate species were included in the multivariate analysis. The selected candidates are shown in the receiver operating characteristics (ROC) curve (Appendix A) and the AUC, best cut off values in Table 1.

Finally, *Streptococcus* (OTU ID: 525966) and *Haemophilus parainfluenza* OTU ID: 4320756) remained as the strongest significant candidates for the HpN group in the age adjusted backward multivariate logistic regression analysis (Table 2).

## 3. Discussion

*H. pylori*-negative gastritis (HpN) is relatively common (1.5–21% of all gastritis) and while it is different and independent from the *H. pylori* infection [5,16] it shares similar at least short-term clinical implications. The etiology of HpN are is not well described [16] and it is possible that other bacterial infections are responsible. In this study, we have described the composition of microbiota in gastric mucosal samples of individuals with HpN compared to two control groups (*H. pylori*-positive gastritis (HpP) and *H. pylori*-negative non-gastritis (control)). The three groups were strictly defined based on multiple diagnostic tests (for the *H. pylori* status) and updated Sydney system (for the gastritis status). Our findings show significant bacterial differences among the three groups that may contribute to the pathogenesis of gastritis, and highlight *Streptococcus* (OTU ID: 525966), *Haemophilus parainfluenza* (OTU ID: 4320756) and *Treponema* (OTU ID: 2707164) as possible unique markers of HpN gastritis. These findings if confirmed may have important clinical implications.

The Kyoto global consensus conference summary acknowledges the specific bacterial etiology-based classification of gastritis not just the *H. pylori* infection but also other bacteria such as *H. heilmannii, Enterococcus, Mycobacteria* and *Syphilis* [17]. Several studies examined gastric mucosal microbiota in *H. pylori*-negative and *H. pylori*-positive patients with gastritis, atrophy and gastric cancer, however, none of these studies considered strict definitions of the *H. pylori* negative status or a histopathological definition of gastritis based on neutrophil infiltration score [11,18]. Strains possess strong stimulatory capacity for neutrophil activation which may play the role in the pathogenesis of gastritis [19]. A recent culture-based study of gastric mucosal tissue showed that *Streptococcus* as well as *Neisseria* were markedly higher in the gastritis group than normal controls [20]. Streptococcal infection was also implicated in the pathogenesis of the rare but fatal phlegmonous gastritis [21]. Furthermore, several studies highlighted that the Streptococcal infection was significantly associated with gastric cancer [3].

The second bacterial candidate for HpN group in our study was *Haemophilus parainfluenzae*, which is an opportunistic pathogen responsible for several infections [22], including the respiratory tract infections [23], endocarditis [24], bacteremia and sepsis [25]. Previous studies reported the detection of lower reads of *Haemophilus parainfluenzae* in the stomach; [26] however it was unclear whether or not it was associated with gastritis. One previous study highlighted that *Haemophilus parainfluenzae* was one of the predominant species among gastric cancer patients based on 16S rRNA amplicon sequencing approach [27]. Importantly, a previous clinical experimental study demonstrated that the *Haemophilus parainfluenzae* bacterial isolation from achlorhydric stomach patients had to increase nitrite accumulation which is a precursor of the carcinogenic N-nitroso compounds that formed in the gastrointestinal tract by a combination of chemical and enzymic reactions, and that the longer it persists, the greater the tendency for the carcinogens to be formed [28].

Lastly, one HpN subject in our study was infected with *Treponema*. Syphilitic gastritis is one of the rare manifestations of syphilis. Our study showed that similar to *H. pylori* infection, *Treponema* dominated other microbiota with a relative abundance of 65% [29].

The gastric microbiome is influenced by other factors that affect gastric acidity such as gastric surgery and PPIs use, which altered gastric microbiota with a shift towards a less healthy microbiome [15,30]. Therefore, a strength of our study is that the results were not confounded by these clinical variables because we excluded patients with antibiotic or acid inhibitors use. The study had few limitations including the limited sample size especially for those with HpN due to the stringent selection criteria used. However, we elected to pursue clearly defined groups and plan to pursue these promising preliminary findings in a larger study. It is also possible that the findings in a Mongolian population are not be generalizable to other populations due to environmental (e.g., diet) or host genetic differences. Lastly, the cross-sectional design precludes causal inferences.

Based on our finding and published literature we hypothesize that pathogenic candidates which are *Streptococcus* and *Neisseria* may play a role for gastric cancer development other than the *H. pylori* infection.

## 4. Materials and Methods

### 4.1. Study Population and Sampling

We conducted a cross-sectional study of subjects prospectively recruited during November 2014 to August 2016 from the Uvs Province (Western part of Mongolia), Khuvsgul Province (Northern), Umnugovi Province (Southern) and Khentii Province (Eastern Mongolia). Study subjects were recruited from volunteers with dyspepsia symptoms through community-based advertisements. We included only those age ≥18 years old and with no history of antibiotic or acid inhibitor use within the prior six months. We excluded subjects with a history of gastrectomy, endoscopic mucosal dissection or *H. pylori* eradication therapy. The gastric biopsy protocol consisted of six specimens: Three from the gastric antrum for microbial examination (one for *H. pylori* rapid urease test, one for *H. pylori* culture and one for microbiome study) and three for histopathological examination (one from the angulus corpus-antrum junction, one from the greater curvature of the corpus and one from the greater curvature of the antrum). The sampling protocol was based on the American Society for Gastrointestinal Endoscopy guideline [31]. Written informed consent was obtained from all participants, and the ethical permission was approved by the Mongolian Ministry of Health (accepted number N3, 2015) Mongolian National University of Medical Sciences (N13-02/1A, 2013), and Oita University Faculty of Medicine (Yufu, Japan) (P-12-10, 2013).

### 4.2. Histological Examination and Gastritis Definition

The gastric biopsy specimens for histological examination were collected in separate tubes containing 10% formaldehyde and kept at room temperature. Serial sections were stained with the hematoxylin eosin and with May–Giemsa stain. An experienced pathologist (T.U.) examined the stained slides and recorded the findings. The degree of neutrophil infiltration was evaluated in antrum, corpus and incisura angulus. The scores were 0 “normal”, 1 “mild”, 2 “moderate”, and 3 “marked” based on the updated Sydney system [32]. A score ≥ 1 was considered as positive for gastritis in any gastric biopsy site.

### 4.3. Determination of H. pylori Infection

Biopsy specimen for the *H. pylori* culture was immediately placed in a dram vial containing 1 mL of cysteine transport medium with 20% glycerol. [33] The collected samples were placed in −20 °C temporarily then kept at −80 °C. Collected samples were transported from Mongolia to Japan with dry ice. *H. pylori* was initially evaluated using five conventional diagnostic tests (culture, histology, immunohistochemistry, serology and rapid urease test). For the *H. pylori* culture, the antral biopsy specimen was homogenized in normal saline solution and placed in a commercially available selective plate (Nissui Pharmaceutical Co. Ltd., Tokyo, Japan). The plates were incubated for up to 10 days at 37 °C under microaerophilic conditions (10% O2, 5% CO2, and 85% N2). *H. pylori* was identified based on colony morphology, Gram staining, and positive reactions for urease tests.

The *H. pylori* load was classified into four grades: 0 “normal”, 1 “mild”, 2 “moderate”, and 3 “marked” based on the updated Sydney system [32]. A load ≥ grade 1 was considered positive for *H. pylori*. Immunohistochemistry commercially available anti-*H. pylori* antibody (Dako, Glostrup, Denmark) was used to confirm the *H. pylori* infection [34]. Serology of the *H. pylori* infection was evaluated using a commercially available anti-*H. pylori* IgG antibody ELISA kit (Eiken Co., Ltd., Tokyo, Japan). The *H. pylori* status by 16S rRNA gene sequencing using a relative abundance cut off value of 2% for positive results was based on a previous study [35].

### 4.4. DNA Extraction and Sequencing

Biopsy specimen for microbiome analysis was collected in 0.5 mL Allprotect Tissue Reagent buffer (QIAGEN, Hilden, Germany). The collected samples were placed in −20 °C then transported from Mongolia to Japan on dry ice. We performed the DNA extraction and purification using the QIAGEN kit according to the manufacturer’s instruction. Amplicon libraries for pair-end (2 × 300 bp) sequencing on the Illumina MiSeq platform (Illumina Inc., San Diego CA, USA) were constructed using universal primers targeted across the V3 and V4 hypervariable regions of the 16S rRNA gene. The 16S rRNA gene was amplified using primers 341F 5′-CCTACGGGNGGCWGCAG-3′ and 805R 5′-GACTACNVGGGTATCTAATCC-3′ which include overhang adapter sequences at the 5′end to add multiplexing indexes. Libraries were cleaned using Agencourt Ampure XP beads according to the manufacturer’s instruction and sequenced on an Illumina MiSeq platform. The bioanalyzer MCE-202 MultiNA system (Shimadzu, Kyoto, Japan) and QuantFluor dsDNA system (Promega Corporation, Madison, WI, USA) were used to examine the quality of sample. All good quality DNA samples were normalized as the same amount.

### 4.5. Sequence Curation and Analysis

Trimming and quality filtering of the 16S rRNA gene sequence data was performed with the CLC genomic workbench 8.5.4 version (QIAGEN). After sequencing for further analysis our criteria for the minimum OTU reads was less than 10 or the minimum relative abundance is less than 0.05% of OTU.

The amplicon sequencing taxonomic and statistical analysis were done by the interactive web-interfaced Calypso software, version 8.72 via http://cgenome.net/calypso/ [36] and R software (R foundation, Vienna, Austria). Alpha diversity was determined by OTU counts, calculating the Shannon and Simpson’s index and assessed by ANOVA. To explore structural differences in the microbial communities among HpN, HpP, and control groups, a beta diversity analysis was conducted using Permutational Multivariate Analysis of Variance (PERMANOVA) to compare groups using the Bray-Curtis distance metrics. The microbial diversity was visualized by DAPC according to the study group. Biomarker discovery analysis was done by LEfSe, and Wilcoxon tests were performed. Identification of co-occurring and mutually exclusive bacteria was performed by the network analysis. Analyses with *p* value < 0.05 were considered statistically significant. *p* value correction included false discovery rate, Bonferroni, or area under curve (AUC) depending on the analysis. Core microbiome analysis was performed based on the presence of taxonomies and its abundance according to comparison groups. The ROC curve analysis was used for selected bacterial biomarkers for *H. pylori*-negative gastritis and species with an AUC in ROC ≥ 0.7 were included in a multivariate backward logistic regression analysis to select final candidate biomarkers.

## 5. Conclusions

In summary, our findings further support the existence of HpN, and suggest its pathogenic bacteria based on the 16s rRNA amplicon sequencing approach at the species level, which are *Streptococcus* (OTU ID: 525966), *Haemophilus parainfluenza* (OTU ID: 4320756) and *Treponema* (OTU ID: 2707164) as the candidate etiologies. A further long-term clinical follow up and confirmatory experimental studies are required to examine whether these infections are causative pathogenic agents from gastritis to gastric cancer patients.

## Figures and Tables

**Figure 1 cancers-11-00504-f001:**
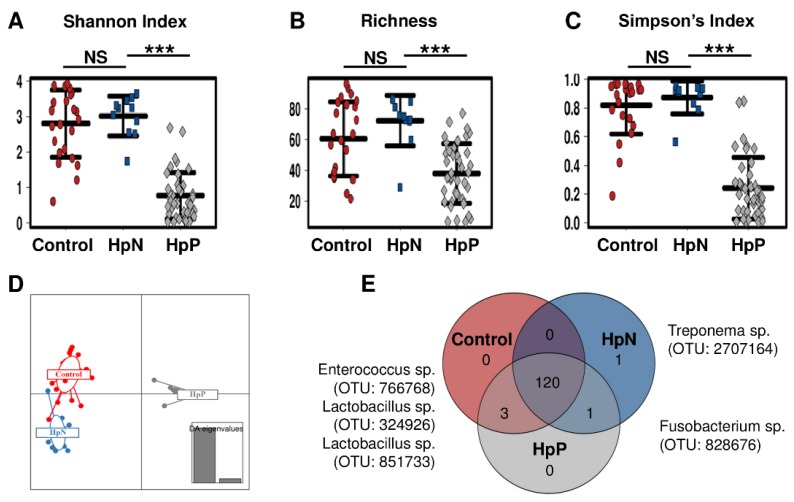
Alpha diversity and unique taxonomy by comparison groups. Shannon index (**A**); bacterial richness (**B**); Simpson’s index (**C**); (**D**) discriminant analysis of principal components (DAPC); (**E**) Venn diagram for overall operational taxonomic units (OTU) taxonomy are shown according to HpN, HpP and control groups. Venn diagram based on the absence or presence of taxonomies showed 120 species were core, four species were pan and one species were determined as unique taxonomy among comparison groups. NS: not significant, *** *p* < 0.001.

**Figure 2 cancers-11-00504-f002:**
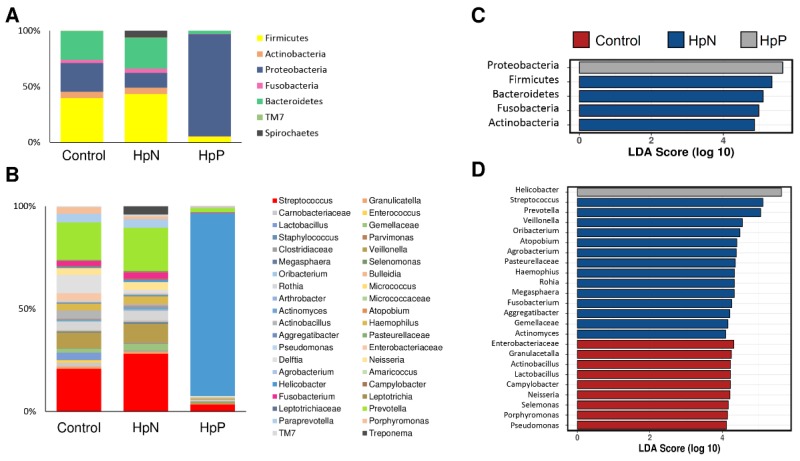
Relative abundance and linear discriminant analysis effect size (LEfSe) by comparison for groups. Microbial relative abundance percentages of the gastric microbiome shown for phylum (**A**) level, genus level (**B**); LEfSe test at phylum level (**C**); at genus level (**D**) are shown according to the HpN, HpP and control groups.

**Figure 3 cancers-11-00504-f003:**
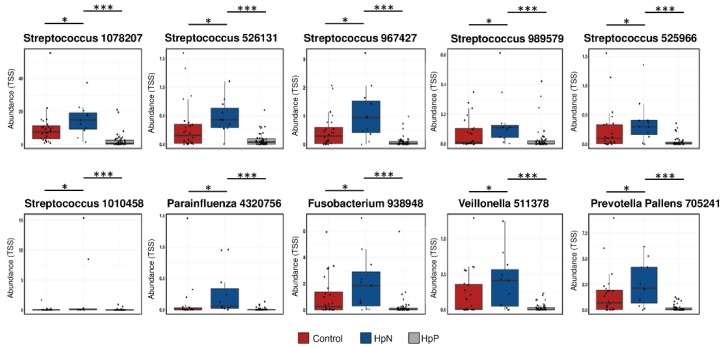
Univariate analysis for selected species according to comparison groups. Each selected species of abundance was shown by the HpN, HpP and control groups. * *p* < 0.05, *** *p* < 0.001.

**Figure 4 cancers-11-00504-f004:**
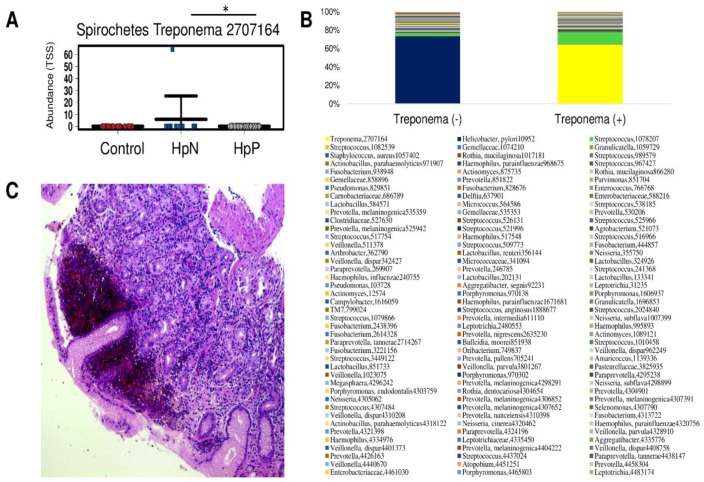
The rare case of syphilitic gastritis and its relative abundance. The mean abundance value of *Treponema* (OUT ID 2707164) at the species level was high in HpN compared with the HpP and control group (**A**); its relative abundance is shown for *Treponema* negative and *Treponema* positive cases (**B**) with its gastric histopathological image (**C**). Inflammatory cell accumulation with hemorrhagic change was observed by hematoxylin and eosin staining in *Treponema* positive patient by 16S rRNA. * *p* < 0.05.

**Table 1 cancers-11-00504-t001:** The best cut off points for selected microbial candidates of *H. pylori*-negative gastritis.

OTU	Area	Lower Bound	Upper Bound	Cut off Value	Sensi-Tivity	Speci-Ficity	*p* Value
*Streptococcus* 1078207	0.74	0.58	0.9	3099	0.64	0.28	0.011
*Streptococcus* 989579	0.68	0.53	0.82			-	NS
*Streptococcus* 967427	0.77	0.6	0.93	138	0.73	0.27	0.005
*Streptococcus* 526131	0.74	0.59	0.88	90.5	0.73	0.27	0.013
*Streptococcus* 1010458	0.73	0.57	0.89	4.5	0.64	0.34	0.015
*Streptococcus* 525966	0.83	0.72	0.94	64.5	0.82	0.19	0.001
*Fusobacterium* 938948	0.73	0.55	0.91	177.5	0.73	0.31	0.014
*Veillonella* 511378	0.78	0.63	0.92	46	0.73	0.27	0.003
*Prevotella pallens* 705241	0.68	0.49	0.88			-	NS
*Haemophilus parainfluenzae* 4320756	0.71	0.54	0.89	6.5	0.73	0.27	0.025
*Helicobacter pylori* 10952	0.75	0.65	0.86	164.5	0.36	0.7	0.008

NS: *p* value is not statistically significant (*p* > 0.05).

**Table 2 cancers-11-00504-t002:** Age adjusted multivariate logistic regression analysis of microbial biomarkers for *H. pylori*-negative gastritis.

OTU	*p* Value	Odds Ratio	Lower 95% C.I.	Upper 95% C.I.
*Streptococcus* 525966	0.009	18.9	2.1	172.8
*Haemophilus parainfluenzae* 4320756	0.025	12.3	1.4	109.6
*Fusobacterium* 938948	NS	7.5	0.6	86.7
*Veillonella* 511378	NS	5.9	0.7	51.4
*Helicobacter pylori* 10952	0.028	0.1	0.0	0.8

NS: *p* value is not statistically significant (*p* > 0.05).

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
