# Peer review of "Gastric Microbiota in Helicobacter pylori-Negative and -Positive Gastritis Among High Incidence of Gastric Cancer Area"

_cancers, 2019, doi:10.3390/cancers11040504_

Reviewer 1 Report

In this manuscript, the authors examined the gastric mucosal microbiota in Hp negative and positive gastritis, even in normal control groups. The authors found lower bacterial richness in Hp positive group. In Hp negative group, they identified several microbiota enrichments. They also proposed several bacterial infection strongly associated with gastritis with Hp negative. In general, the manuscript has no much novelty because quite a lot papers have reported the microbiota distribution in gastritis. It is well known that the negative correlation of microbiota abundance and Hp positive. However, the authors identified some microbiome correlated with gastritis occurrence and this point seems very interesting.

Major concerns:

1, I think the most important issue is the sample size. As we know the diversity of microbiota in quite different cases, it is hard to draw such a solid conclusion that some specific microbiome is significantly is correlated with gastritis occurrence. Nor the authors employed animal model to support their findings.

2, Can the authors draw ROC curve to check the performance of each microbiome in the differentiation of Hp negative and positive cases or the performance of a microbiota combination?

3, Do the authors have follow up data to check the microbiota changes before and after treatment in Hp negative and positive cases respectively?

Author Response

Responses for comments by Reviewer 1:

Q1: I think the most important issue is the sample size. As we know the diversity of microbiota in quite different cases, it is hard to draw such a solid conclusion that some specific microbiome is significantly is correlated with gastritis occurrence. Nor the authors employed animal model to support their findings.

Response: Thank you very much for your comment. We addressed these points as the study limitation in the discussion part on page 8, L248-253. In addition, animal model should be done in the future study and we are now planning to perform the experiments.

Q2: Can the authors draw ROC curve to check the performance of each microbiome in the differentiation of Hp negative and positive cases or the performance of a microbiota combination?

Response: Thank you very for your comment. We checked the ROC curve for H. pylori-negative gastritis vs control group (H. pylori-positive gastritis and non-gastritis group). The area under curves, p values, the best cut-off points, sensitivity and specificity of each candidate OTUs are summarized in Table 1 on page 6, L196-197. 

Q3. Do the authors have follow up data to check the microbiota changes before and after treatment in Hp negative and positive cases respectively?

Response: Unfortunately, we did not follow up those patients after treatment. We will plan to do the experiments in the near future.

Reviewer 2 Report

This is important description though preliminary and descriptive in nature.

The author should add IRB approval number. And according to the population studied, all the participants had dyspeptic symptoms, but the diagnoses of gastritis were not made in "Normal" control, which was used for this investigation. Clarify this point.

Have the authors ever done non-gastiris (definition is not easy for the reality when including parasites or viral gastritis; symptoms are very subjective) material such as autopsy stomach without any gastric disease.

How about gastric cancer cases which may have gastritis surrounding the cancer?

Author Response

Responses for comments by Reviewer  2:

Q1: The author should add IRB approval number.

Response: Thank you very much for the precious advice. We added IRB on page 8, L267-270.

Q2: According to the population studied, all the participants had dyspeptic symptoms, but the diagnoses of gastritis were not made in "Normal" control, which was used for this investigation. Clarify this point.

Response: Thank you very much. Study participants had dyspeptic symptoms, however disease status based on histological examination included normal control group. The criteria for normal control group was as follows: Updated Sydney System score for neutrophil infiltration score 0 (none of infiltration), and negative by all H. pylori tests including 16S rRNA gene amplicon sequencing. However these patients could be have the functional disorder such as dysphagia or reflux. We clarified this on page 8, L276-277.

Q3: Have the authors ever done non-gastritis (definition is not easy for the reality when including parasites or viral gastritis; symptoms are very subjective) material such as autopsy stomach without any gastric disease.

Response: Thank you very much for your comments. We agree that the definition of gastritis is not easy. In our study, the definition of gastritis is based on histological examination (neutrophil infiltration score based on Updated Sydney System score). All participants were examined by pathological confirmation. We did not include the pure healthy peoples since these peoples are rarely enrolled to the endoscopic examination. However among our study participants we had non-gastritis patients based on our criteria that there is no neutrophil infiltration seen in the gastric mucosa (Updated Sydney System score 0).

Q4: How about gastric cancer cases which may have gastritis surrounding the cancer?

Response: It could be probably, in this study we did not examine.

Reviewer 3 Report

This study validated the gastric flora of patients with H. pylori-positive and negative gastrit and of non patients to reveal microbiome changes associated with disease state. In recent years, it has become clear that various diseases are associated with abnormalities of the intestinal bacterial flora. This undesirable intestinal flora is called dysbiosis. H. pylori infection is commonly etiologically associated with peptic ulcer and gastric cancer. But now, it is raised as question whether the concept that most stomachs could be characterized as either normal or H. pylori-infection might be too simple. The disease concept of H. pylori negative gastritis is still not well established. Therefore, this research result has good novelty and its value is considered to be enough.

Minor questions

The numbers of sequence reads from some samples are only 2000, which is significantly less than others (supplementary figure 2). It seems that some of these values are too small to cover all species from the analysis by the rarefaction curves (supplementary figure 3). The authors should explain the adequacy of evaluation for such samples in the same way as others.

Page 3, L95, none of closing parenthesis ")"

Page 6, L184, typo "Trepenoma"

Author Response

Responses for comments by Reviewer 3:

Q1: The numbers of sequence reads from some samples are only 2000, which is significantly less than others (supplementary figure 2). It seems that some of these values are too small to cover all species from the analysis by the rarefaction curves (supplementary figure 3). The authors should explain the adequacy of evaluation for such samples in the same way as others.

Response: Thank you very much for your precious comment. The bioanalyzer MCE-202 MultiNA system (Shimadzu, Kyoto, Japan) and QuantFluor dsDNA system (Promega Corporation, Madison WI, USA) were used to examine the quality of sample. All good quality DNA samples were normalized as the same amount. After sequencing for further analysis our criteria for the minimum OTU reads was less than 10 or the minimum relative abundance is less than 0.05% of OTU. Therefore we assume that the cases with around 2000 overall reads are still adequate for the analysis. We added this detailed protocol in methods part on page 9, L313-315 and L318-319.

Q2: Page 3, L95, none of closing parenthesis ")"

Response: Thank you very much. We added closing “)” marks on page 3, L97.

Q3: Page 6, L184, typo "Trepenoma"

Response: Thank you very much. We corrected as “Treponema” on page 6, L185.
